# Contact lenses contamination by *Acanthamoeba* spp. in Upper Egypt

**Faten A. M. Hassan**[1¤]*, **M. E. M. Tolba**[2], **Gamal H. Abed**[3], **H. M. Omar**[4], **Sara S. Abdel-Hakeem**[3]

**1** Faculty of Science, Department of Microbiology, Taiz University, Taiz, Yemen, **2** Faculty of Medicine, Department of Parasitology, Assiut University, Assiut, Egypt, **3** Faculty of Science, Department of Zoology, Parasitology laboratory, Assiut University, Assiut, Egypt, **4** Faculty of Science, Department of Zoology, Physiology laboratory, Assiut University, Assiut, Egypt

¤ Current address: Faculty of Science, Department of Zoology, Assiut University, Assiut, Egypt
* fatenhassan28@yahoo.com

**Data Availability Statement:** All relevant data are within the paper and its Supporting Information files.

## Abstract

### Background

*Acanthamoeba* spp. are one of the free-living amoeba that spread worldwide causing keratitis. Owing to the increase in the use of lenses, whether for medical or cosmetic purposes, the incidence of disease increases every year. Contamination of the lenses with the *Acanthamoeba* trophozoites or cysts may lead to eye infection and cause sight-threatening keratitis in human. We isolated *Acanthamoeba* spp. from new lenses, used lenses, and contact lens disinfecting solutions and identified them based on morphological characteristics and molecular test.

### Methods

New and used lenses and contact lens disinfecting solutions were cultured on monogenic media. Light and scanning electron microscope was used to identify *Acanthamoeba* spp. morphological features. Genotype identification was also evaluated using PCR sequencing of 18S rRNA gene specific primer pair JDP1 and JDP2.

### Results

A hundred samples were examined, 29 (29%) were infected with *Acanthamoeba* spp. That belonged to two strains of *Acanthamoeba* (*Acanthamoeba* 41 and *Acanthamoeba* 68). 18S rRNA of the *Acanthamoeba* 41 had 99.69% sequence identity to *Acanthamoeba castellanii* clone HDU-JUMS-2, whereas *Acanthamoeba* 68 had 99.74% similar pattern to that of *Acanthamoeba* sp. isolate T4 clone ac2t4 that are morphologically identified as *Acanthamoeba polyphaga*. The obtained data revealed that the isolated strains belong to T4 genotype that was evolutionarily similar to strains isolated in Iran.

### Conclusions

Cosmetic lenses and disinfectant solutions are a major transmissible mode for infection. This genotype is common as the cause of *Acanthamoeba* keratitis. To avoid infection, care

**Funding:** The author(s) received no specific funding for this work.

**Competing interests:** The authors have declared that no competing interests exist.

must be taken to clean the lenses and their preservative solutions and prevent contamination with the parasite.

## Introduction

The use of contact lenses has become a daily habit for a large percentage of people. Approximately 140 million individuals wear contact lenses worldwide for optical, occupational, and cosmetic purposes [1]. However, many studies reported contact lens–related eye infections, because some contact lens wearers had developed microbial keratitis that can lead to serious outcomes, including blindness. In 2015, the Centers for Disease Control and Prevention reported adult contact lens wearers in the United States who were at risk of contact lens–related eye infections [2].

Acanthamoeba keratitis (AK) is a corneal infection caused by *Acanthamoeba*, which is a type of free-living amoeba (FLA). It can be found in various water sources such as brackish water, seawater, groundwater, drinking water, river water, wastewater, and pool water and potentially contaminated cleaning solutions of contact lenses. It can also grow in contact lenses that are cleaned with contaminated tap water [3].

Contamination of contact lens storage systems, poor contact lens hygiene, ineffective contact lens disinfecting solutions, or contact with contaminated water and mud can cause keratitis infection [4,5]. It is most common in contact lens wearers. Contact lens users comprise >85% of patients with AK who attributed to the popularization of lenses [5]. Jones [6] reported the first case of *Acanthamoeba* infection in a contact lens wearer, with reported rates in the range of 1 to 33 cases per million of contact lens wearers per year [7]. During its life cycle, *Acanthamoeba* spp. have two stages, a metabolically active and more susceptible trophozoite form and a dormant, highly resistant cyst form. The trophozoites and cysts attach to the surface of contact lenses and are transmitted to the eye [5]. Approximately 30 species of *Acanthamoeba* are categorized into three morphological groups based largely on the cyst morphology of the species including endocysts, ectocysts, and their size [7,8]. The species that have been reported as causes of eye infections in contact lenses wearers are *Acanthamoeba castellanii*, *Acanthamoeba polyphaga*, *Acanthamoeba rhysodes*, *Acanthamoeba culbertsoni*, *Acanthamoeba lugdunensis*, *Acanthamoeba griffini*, *Acanthamoeba hatchetti*, *Acanthamoeba quina*, *Acanthamoeba lenticulata*, and *Acanthamoeba triangularis* [9].

Recent studies have been able to group these species based on the sequence of their genomic DNA. Because of its more sensitivity and specificity, conventional polymerase chain reaction (PCR) has been developed as a reliable method to confirm the detection of free-living amoeba in environmental samples and in clinical samples. This method has proved to be a more effective way of identifying various species than traditional parasitological techniques using morphological features because the morphological features use has been reported as problematic owing to inconsistencies and variants of the cyst morphology [10].

Previous studies identified 20 genotypes of *Acanthamoeba*: T1–T20 by molecular approach using SSU18S rDNA gene segments [11]. Recent epidemiological researchers found that the T4 genotype group of *Acanthamoeba* is the most abundant in the environment and includes many pathogenic strains that have been associated with AK [12–14]. Therefore, this study aimed to examine the rate of contamination of new and used contact lenses and disinfectant solutions with *Acanthamoeba* and conduct morphological and molecular genotyping on isolates in Upper Egypt.

## Materials and methods

### Ethics statement

The need of consent was waived by the research ethics committee, Faculty of Science, Assiut University, as the participants' voluntary donated the used lenses for sake of scientific research. While the new ones were purchased by the authors as commercially available goods.

### Collection and cultivation of samples

From April to August 2019, 100 samples of cosmetic (colored, not intended for medical use) lenses and contact lenses solutions, new lenses (NL, n = 50), used lenses (UL, n = 30), and contact lenses solutions (used) (LS, n = 20) were collected from cosmetic shops in Assiut Governorate, Egypt, and from contact lenses users. Each sample was placed immediately onto the 1.5% non-nutrient agar (NNA) medium plates containing 0.12g NaCl, 0.004g $MgSO_4.7H_2O$, 0.004g $CaCl_2.2H_2O$, 0.142g $Na_2HPO_4$, 0.136g $KH_2PO_4$, and 15.0g agar/L of distilled water at pH 6.8 [15]. They were seeded with live *Escherichia coli* (*E. coli*) ATCC 25922 (ANNE). The plates were then incubated at 30°C under standard atmospheric conditions, and *Acanthamoeba* growth was monitored daily for 7 days under an inverted microscope for trophozoites and for 14 days for cysts [16].

### Morphological identification and staining techniques

10 mL of sodium phosphate buffer (pH 7.4) were added to a newly sub cultivated NNA- *E. coli* with amoebae plate and pipetted at the 5th to 6th days to remove the trophozoites and on the 10th to 14th days to remove cysts from the agar surface. To minimize the presence of *E. coli*, the mixtures were centrifuged three times at 600 xg for 10 minutes at room temperature. The supernatant was discarded and the sediment was used for further microscopic examination and staining procedures [17]. Furthermore, isolated members belonging to *Acanthamoeba* were morphologically identified to the species level [18].

Aqueous solutions of 0.2% iodine, 0.1% eosin, and 0.1% methylene blue stains were used in wet mount for temporary staining [13]. Giemsa stain (GS) (stock solution 1: 5 in buffered water, pH 7.2) was used for a permanent stained smear. Smears were examined under light microscopy (Olympus, Japan) [17].

### Scanning electron microscopy

Representative specimens from *Acanthamoeba* spp. were suspended in 0.1 M sodium phosphate buffer (pH 7.4), centrifuged at 600×g or 5 minutes to remove the remaining mediums, and washed with 0.1 M sodium phosphate buffer (pH 7.4) at room temperature. Then, pellets of both trophozoites and cysts were fixed in a mixture of 2.5% paraformaldehyde and 5% glutaraldehyde in 0.1 M sodium phosphate buffer, pH 7.3, at 4°C for 24 hours. Thereafter, the specimens were washed three times for 5 minutes in the fixative buffer, post fixed in 1% osmic tetraoxide at 4°C for 1 hour, and washed twice in 0.1M sodium phosphate buffer. The samples were dehydrated using ascending concentrations of ethyl alcohol and maintained in isoamyl acetate for two days and subjected to critical-point drying with a Polaron apparatus. Finally, the samples were mounted on aluminum stubs and coated with gold using JEOL 1100 E ion sputtering device and observed with a JEOL scanning electron microscope (SEM) (JSM 5400 LV) at 10 KV [19] in the Electron Microscope Unit in Assiut University, Egypt.

## DNA extraction and amplification with polymerase chain reaction

The isolates species that were morphologically identified of *Acanthamoeba* spp. were subjected to molecular characterization at the genus level. Cysts were harvested and centrifuged at 600 xg for 5 minutes. The supernatants were discarded, and the pellets were suspended in phosphate-buffered solution to a final volume 200 μL and boiled at 80˚C for 1 hour. The DNA was extracted using the QIAGEN extraction kit (QIAamp® DNA Minikit, Hilden Germany) following the manufacturer's protocol in the Molecular Biological Unit in Assiut University. The DNA concentration was determined using a Nano Drop spectrophotometer (Fisher Scientific) and stored at −20˚C for further PCR analysis. Based on the morphological identification, genus-specific primers JDP1 and JDP2 were purchased from *Thermo Fisher Scientific in United States* and used in PCR to amplify the most informative region of 18S ribosomal (r) RNA (ASA.S1) of genus *Acanthamoeba* [20].

The sequence of forward and reverse primers was (GDP1: F5'–GGC CCAGATCGTTTAC CGTGAA-3') and (GDP2:R5' TCTCACAAGCTGCTAGGG AGTCA-3'), respectively. Amplification was performed in a 25μL volume containing 1 μL template DNA extract, 12.5μL TaqTM Red Mix (Bioline USA Inc., Boston, USA), 1 μL forward primer, 1 μL reverse primer, 0.5 μL Taq DNA polymerase (Bioline, USA), and 9 μL DNA-free water. A Veriti ™ 96-well thermal cycler (9902, Singapore) was used for 40 cycles as follows: The reaction was performed at 94˚C for 5 minutes, followed by 40 cycles at 94˚C for 1 minute, 60˚C for 1 minute, 72˚C for 1 minute, and an extension at 72˚C for 5 minutes. The control specimen was conducted using a template DNA-free blank (Schroeder *et al*., 2001). Aliquots of 10 μL from each PCR reaction was subjected to a 1.5% agarose gel in horizontal cell (Compact M, Biometric, Germany) and stained with ethidium bromide, and DNA fragments were observed under ultraviolet illumination. The size of each fragment was based on a comparison with 100-bp ladder.

## Sequencing and phylogenetic analysis

The purified PCR product was then sent to SolGent Company, Daejeon, South Korea, for 18S gene sequencing using GDP1 forward and GDP2 reverse primers. The obtained sequences were compared with those of *Acanthamoeba* sequences in GeneBank using Basic Local Alignment Search Tool engine from the National Center of Biotechnology Information website. The phylogenetic analysis was performed DNASTRA MegAlgin software 5.01@DNA. Cluster X and GeneDoc were used to determine the alignment and percentage of sequence dissimilarity. The phylogenetic tree was generated with cluster W method using Kimura 2-parameter algorithm with bootstrap analysis of 1000 replicates with *Balamuthia mandrillaris* and *Plasmodium falciparum* as out-group for construction of phylogenetic tree [21].

## Results

### Frequencies of infection

Of the 100 samples, 33 (33%) culture plates were positive with Vahlkampfiidae *(Naegleria)* and *Acanthamoeba* spp., 18 (36%) in NL, 4 (13.3%) in UL, and 11 (55%) in LS (Table 1). The Vahlkampfiidae *(Naegleria)* and *Acanthamoeba* spp. were identified in the positive samples according to the morphological criteria of cysts and trophozoites.

### Light microscopic structure of *Acanthamoeba* spp. trophozoites and cysts

Trophozoites and cysts isolates exhibited morphological characteristics of two different species that are typical of group II species. Under light microscope, the trophozoite size was approximately 20 to 26 μm long with many spine-like processes called acanthopodia. The ectoplasm

**Table 1. Presence of *Acanthamoeba*, free living amoeba, and mixed genera in the examined samples.**

| Parasite spp. | NL (n = 50) | UL (n = 30) | LS (n = 20) | Total (n = 100) |
|---|---|---|---|---|
| *Acanthamoeba* spp. | 15 (30%) | 4 (13.3%) | 10 (45%) | 29 (29%) |
| **Vahlkampfiidae** (*Naegleria*) | 2 (4%) | — | — | 2 (2%) |
| **Mixed** | 1 (2%) | — | 1 (5%) | 2 (2%) |
| **Total** | 18 (36%) | 4 (13.3%) | 11(55%) | 33 (33%) |

was clear, and the endoplasm was finely granulated with a spherical nucleus and an obvious well-defined vacuole (Figs 1A and 2A). The cysts were round, oval, and sometimes slightly deformed, with a range of 10 to 20 μm in diameter, and uninucleate, and the number of angles was 2 to 5. The outer cyst wall, the ectocyst, was wrinkled and the inner cyst wall, the endocyst, was polygonal with opercula. The ectocyst was conspicuously separated from the endocyst by a lucent intercyst space except in the region of indistinctive cyst pores (ostioles) (Figs 1B and 2B).

The ultrastructure of the trophozoites and cysts of *Acanthamoeba* isolated strain under SEM showed the characteristics of trophozoite with prominent and numerous needle-like structures, whereas the cysts appeared wrinkled and had thick ridges over their entire surface, giving a pitted and cyst showing ostioles that connect the endocyst and ectocyst layers. Trophozoites and cysts were surrounded by bacteria (Figs 3 and 4).

Trophozoites and cysts gave variable results with different stains to clarify the morphological details. The iodine wet mount stain showed a yellowish-brown color reaction to

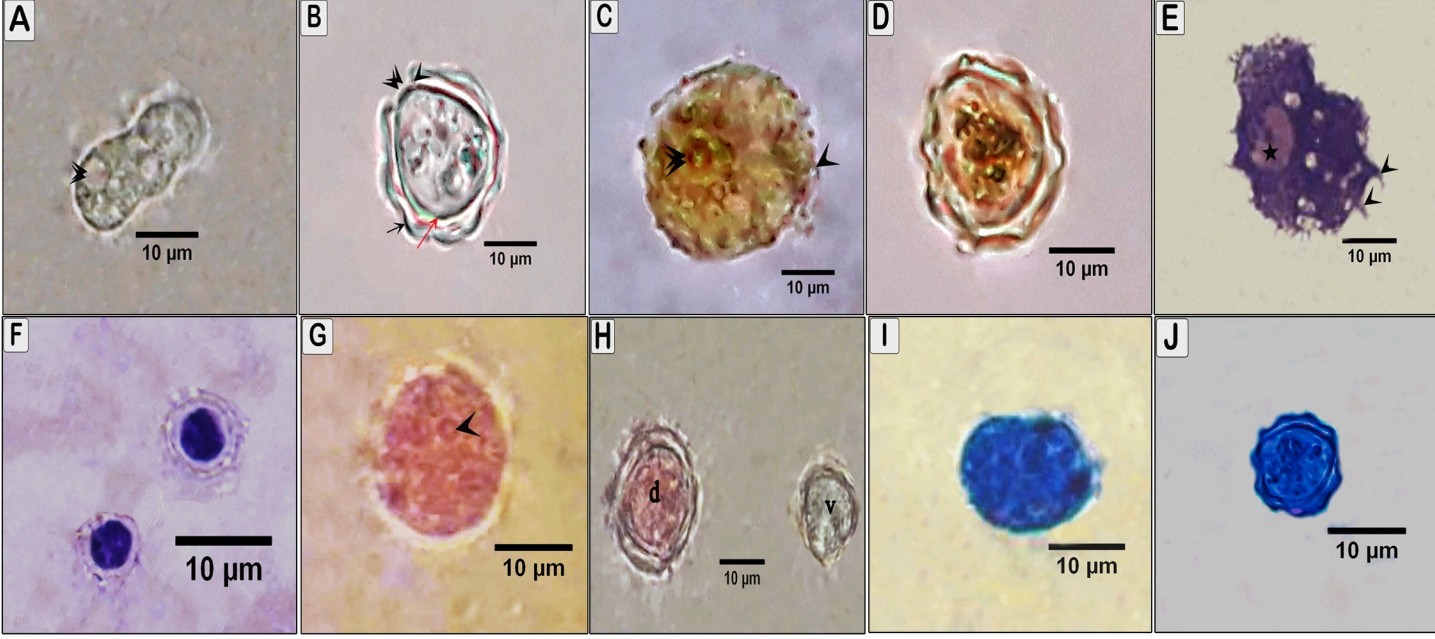

**Fig 1. Light micrographs of *Acanthamoeba castellanii*, trophozoite and cyst staining with different stains (X1000).** (A) *A. castellanii* trophozoite in wet mount showing vacuoles (arrowhead). (B) Unstained *Acanthamoeba* cysts showing wrinkled ectocyst (black arrow), smooth endocyst (red arrow) and nucleus and ostiole (arrowhead). (C) *A. castellanii* trophozoite stained with iodine wet mount stain trophozoite appears yellowish–brown with prominent nucleus (arrowhead). (D) *A. castellanii* cyst appears yellowish–brown with well-defined ectocyst and endocyst. (E) *A. castellanii* trophozoite stained with Giemsa stain showing prominent nucleus (star) contractile vacuole and acanthopodia (arrowhead). (F) The nucleus of cyst stained blue with Giemsa stain. (G) *A. castellanii* stained with eosin showing dead trophozoite appears pink color with prominent nucleus (arrowhead). (H) *A. castellanii* cyst showing dead cyst appears pink (d) and viable cyst unstained (v). (I) *Acanthamoeba* trophozoite and (J) cyst stained with methylene blue.

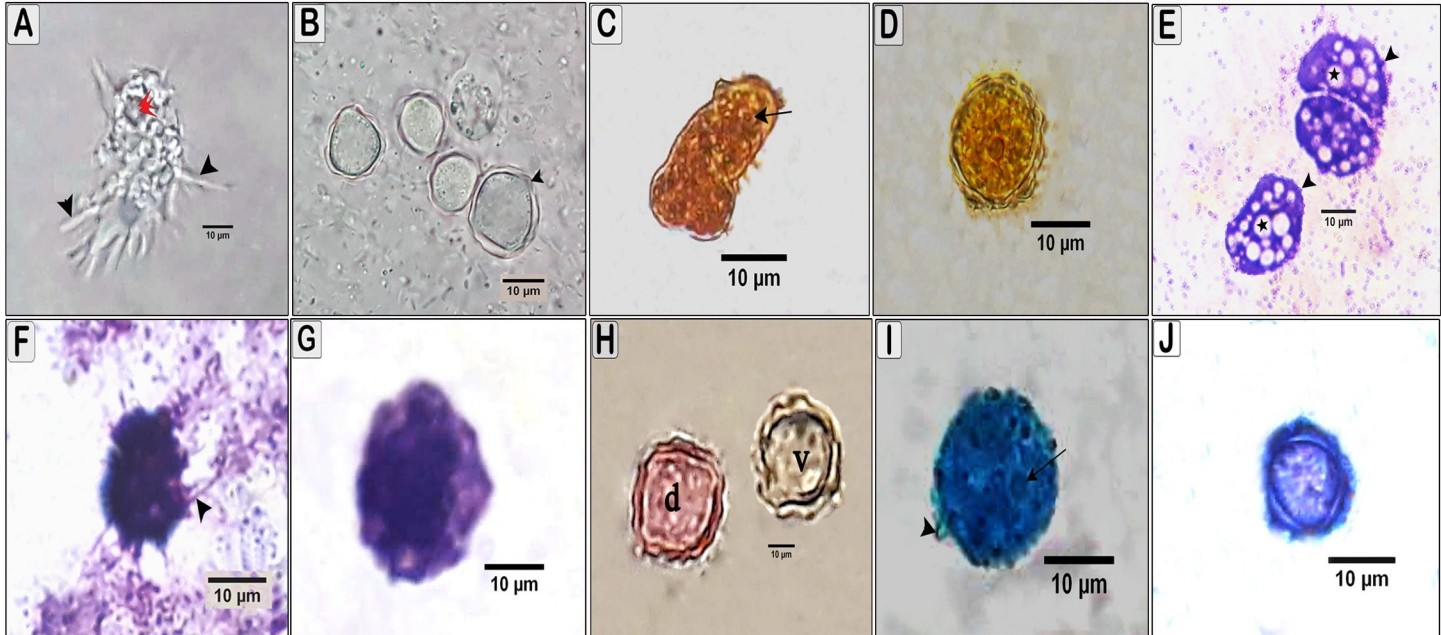

**Fig 2. Light micrographs of *Acanthamoeba polyphaga* trophozoite and cyst staining with different stains.** (A) *A. polyphaga* trophozoite in wet mount showing numbers of spiky projecting acanthopodia (arrowhead) and vacuoles (red arrow head) X400. (B) Unstained *A. polyphaga* cysts showing wrinkled ectocyst (arrowhead), smooth endocyst and nucleus X400. (C and D) *A. polyphaga* trophozoite and cyst stained with iodine wet mount stain appeared yellowish–brown with well-defined ectocyst and endocyst and central nucleus X1000. (E and F) *A. polyphaga* trophozoite stained with Giemsa showing prominent contractile vacuole (star) and acanthopodia (arrowhead) X1000. (G) *A. polyphaga* cyst stained with Giemsa X1000. (H) *A. polyphaga* with eosin showing dead cyst appears pink (d) and viable cyst unstained (v). (I and J) *A. polyphaga* trophozoite and cyst stained with methylene blue. They appear blue and the nucleus of trophozoite appears dark blue (arrow) and acanthopodia (arrowhead) X1000.

trophozoite and cyst (Figs 1C, 1D, 2C and 2D). Giemsa stain (Figs 1E, 1F and 2E-2G). The dead trophozoites and cysts appear reddish to pink color with well-defined well differentiation, whereas viable ones appear unstained (Figs 1G, 1H and 2H). Methylene blue (Figs 1I, 1J, 2I and 2J) gave poor visibility of *Acanthamoeba* cyst, but with a good visibility of trophozoites.

## 18S rRNA sequence analysis of the two isolated Acanthamoeba strains

*Acanthamoeba* spp. was identified from 29 of 100 (29%) positive culture plates according to characteristic morphology. Positive isolates were tested using PCR, and a genus-specific primer pair for *Acanthamoeba* spp. The full lengths of the 18S rRNA genes of the *Acanthamoeba* sp. isolated strains *Acanthamoeba* 41 and *Acanthamoeba* 68 were 345 and 405 bp, respectively. *Acanthamoeba* 41 had very high 18S rDNA sequence similarity with *A. castellanii* clone HDU-JUMS-2 (99.42%), whereas *Acanthamoeba* 68 had very high 18S rDNA sequence similarity with *Acanthamoeba* sp. isolate T4 clone ac2t4 (99.74%), which are morphologically close to *Acanthamoeba polyphaga*. Sequencing alignment revealed that 18S rDNA sequences of *Acanthamoeba* spp. strains *Acanthamoeba* 41 and *Acanthamoeba* 68 corresponded to genotype T4 (Figs 5 and 6).

## Discussion

Several authors reported that the continuous increase in the use of contact lenses worldwide was correlated with the increase in keratitis infections. This study was designed to detect the presence of FLA, especially *Acanthamoeba* spp., which is the common parasitic causative of keratitis, in commercial cosmetic lens, particularly colored lens and disinfectant solutions sold

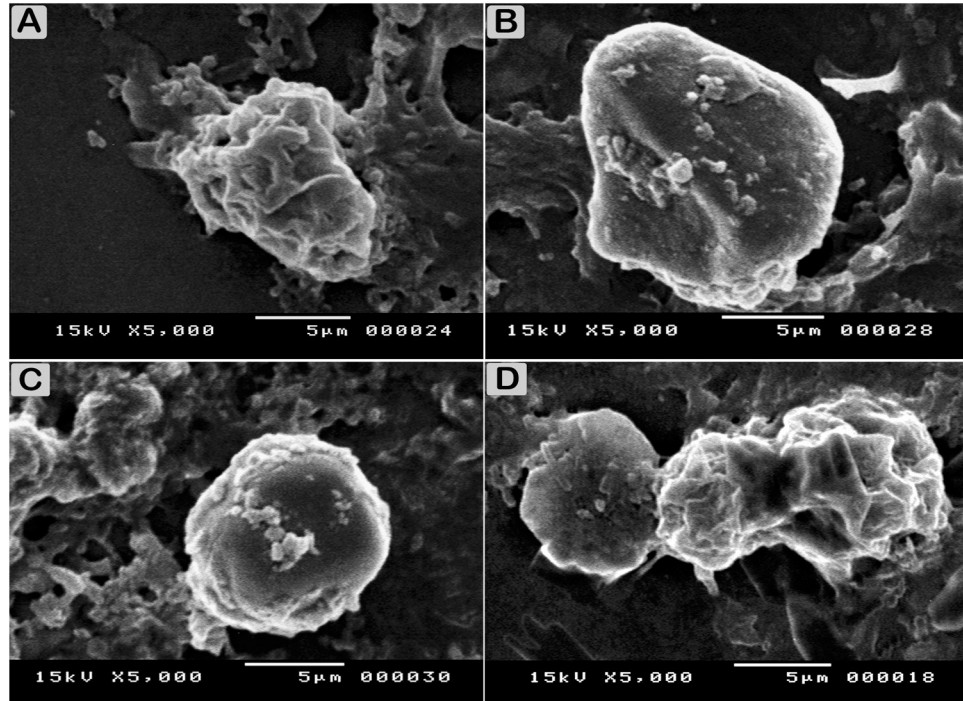

**Fig 3. Scanning electron micrograph showing morphological stages of *A. castellanii*.** (A) Trophozoite with the characteristic acanthopodia. (B and C) Phases of cyst wall formation as detected by scanning electron microscopy showing precyst with cellulose patches on the cell surface. (D) Mature cyst, the cyst wall appears completely formed and the wrinkled exocyst is clearly detectible.

and used in Assiut Governorate, Egypt, by culturing of samples, morphological identification, and molecular confirmation.

In this study, *Acanthamoeba* was presented in 30%, 13.3%, and 45% of NL, UL, and LS, respectively. In a 10-year survey (1994–2004), Ibrahim *et al.* [22] reported that *Acanthamoeba* was isolated in contact lenses and contact lens disinfecting solutions in all cases of AK. The presence of *Acanthamoeba* in cosmetic CLs may refer to irregular and rougher colored surface of cosmetic CLs which could facilitate *Acanthamoeba* adhesion on the surface through acanthopodia [23]. In addition, contact lens and contact lens disinfecting solutions may be contaminated with *Acanthamoeba* from tap water or dust [24].

This study found that the trophozoite that was developed from a cyst shell and nuclei at day 4 was irregular in shape and the cytoplasm contained a central nucleus with contractile vacuoles and has thorn with tapered end known as acanthopodia. This finding was in agreement with Muslim and Azhar *et al.* [25]. In addition, cysts of *Acanthamoeba* spp. appeared with double cell wall, and the ectocyst appeared wrinkled, which was consistent with Sampaotong *et al.* [26] who found that an ectocyst appears wrinkled and clearly separated from the endocyst that was thin and smooth. The cyst and trophozoite morphology characterized in the present study resembled to the various species of group II based on size, morphology, and number of opercula [4].

Wet mount preparation has the advantage of demonstrating the trophozoite motility. However, the internal structures are often poorly visible making the definitive diagnosis of cysts or trophozoites difficult leading to misdiagnosis of *Acanthamoeba* in 60% to 70% of AK cases [17]. Therefore, several staining techniques were used in this study to easily identify

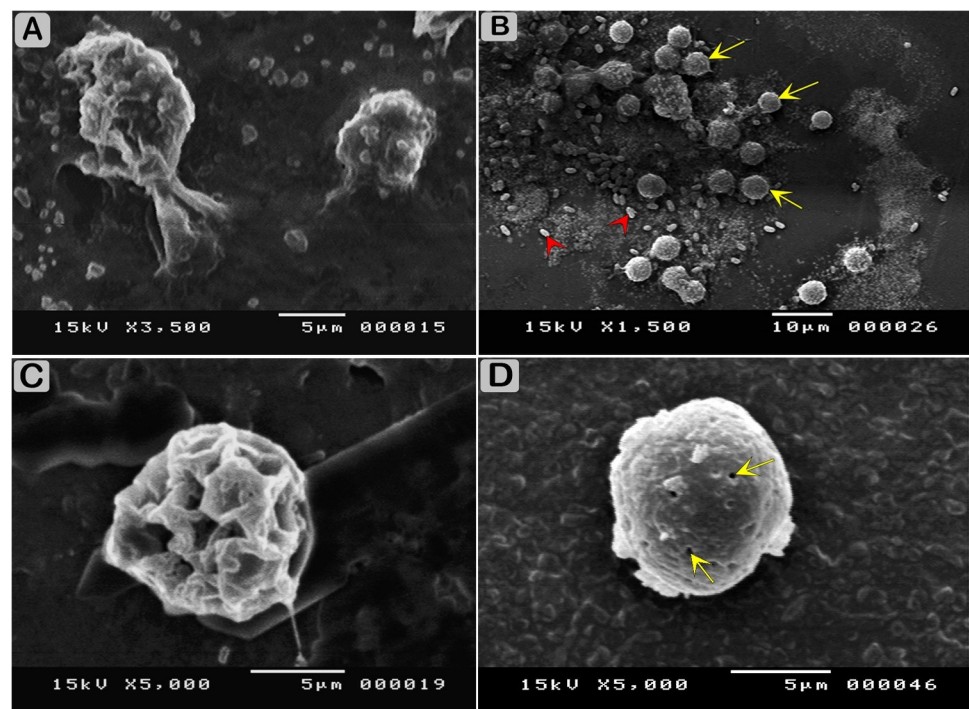

**Fig 4. Scanning electron micrographs of *A. polyphaga* showing morphological stages.** (A) Trophozoite of *A. polyphaga* showing irregular cell shape and acanthopodia structure. (B) SEM showing numerous cysts (yellow arrows) and bacteria cells (red arrow heads). (C) SEM of *A. polyphaga* cyst showing the typically wrinkle appearance with high ridges over the surface. (D) SEM of *Acanthamoeba* cyst showing ostioles (yellow arrows) that connect the endocyst and ectocyst layers.

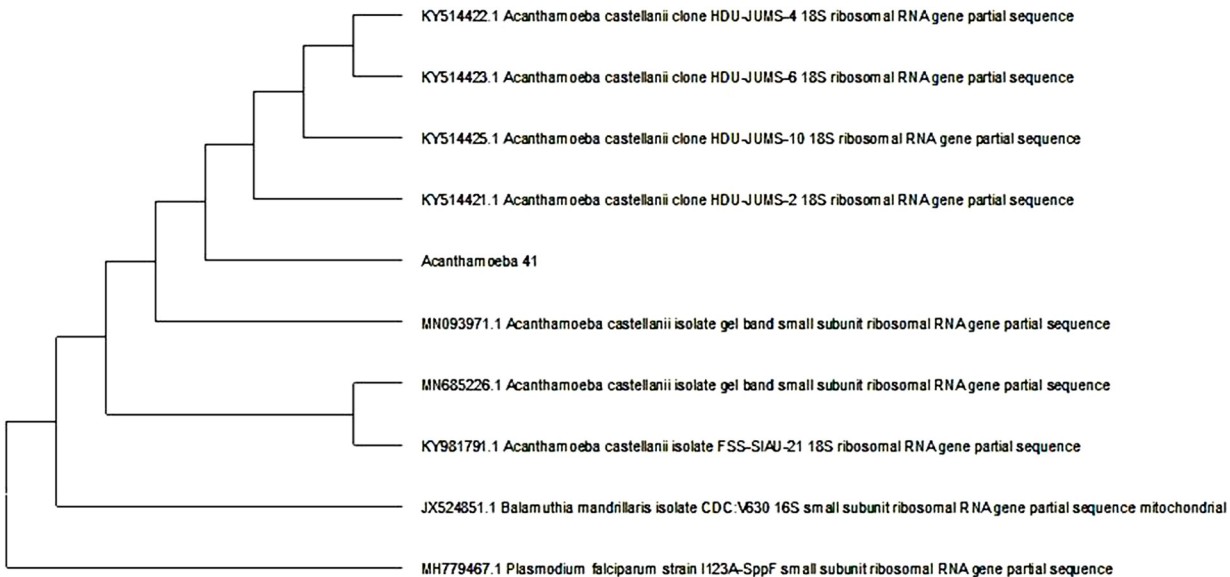

**Fig 5. Phylogenetic tree derived by aligning the 18s rDNA sequence of the strain isolate (41).** The results obtained showed that its 18s rDNA sequence shared considerable homology with *A. castellanii*.

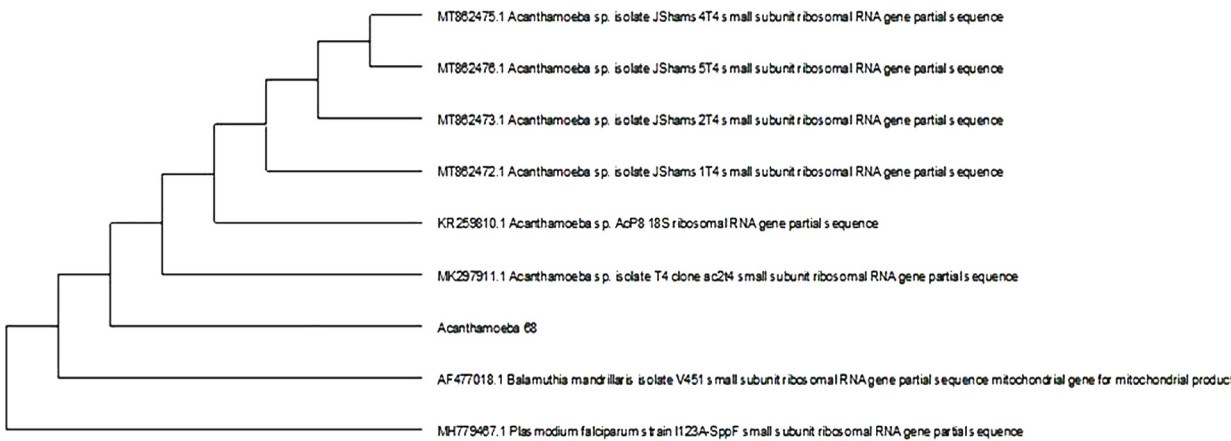

**Fig 6. Phylogenetic tree derived by aligning the 18s rDNA sequence of the strain isolate (68).** The results obtained showed that its 18s rDNA sequence shared considerable homology with *A. polyphaga*.

*Acanthamoeba* cysts. It was found that stained preparations gave variable results and the stain type, the concentration of the staining solution, temperature, and duration of staining affect the staining quality of *Acanthamoeba* cysts.

Some staining techniques were recommended in experimental conditions and clinical specimens to detect the morphological details of the different phases and facilitate the identification of the isolated strain [13,27]. In the current study, Giemsa and methylene blue stains were found to be effective to visualize the morphological details of *Acanthamoeba* trophozoites, whereas iodine was effective for staining *Acanthamoeba* cysts. These results correspond to the data obtained by Muchesa *et al.* [28] and Behera and Satpathy [29]. Contradictory results were described by Garajová *et al.* [30] who showed that *Acanthamoeba lugdunensis* and *Acanthamoeba quina* cysts had no color reaction when staining with iodine. El-Sayed and Hikal [17] showed no contrast in the *Acanthamoeba* spp. stained with methylene blue. Variable results with stains between *Acanthamoeba* spp. depend on many factors such as the fixation of specimens, concentration of the staining solution, duration of the staining, quality of the culture strain, and the temperature.

In this study, PCR assay based on sequence analysis of the 18S rRNA gene and a phylogenetic tree revealed that the sequencing identifying of the two selected strains, *Acanthamoeba* 41 and *Acanthamoeba* 68, isolated from the commercial lens and disinfectant solutions, in Assiut, Egypt, which were classified in group II according to their morphology characters were positioned close to genotype T4 species that were evolutionarily related to strains isolated in Iran [31,32]. Aghajani *et al.* [33] reported that the increasing importance of T4 genotype group refers to its high range of distribution, resistance of its cysts to antiseptics, and its production of more cytotoxic factors than other genotypes. Moreover, Taher *et al.* [34] reported that genotype T4 was the most prevalent in corneal infection with an evolutionary lineage associated with keratitis in Egyptian contact lens users. Genotype T4 also includes 15 strains with sequence differences ranging from 0% to 4% [35].

Despite many reports on the prevalence of *Acanthamoeba* in environmental [36–38] and clinical [34] isolates, this report is the first data on the pathogenic *Acanthamoeba* frequency and morphological and molecular description of FLA isolated from new commercial lenses and preserved lens solution in Upper Egypt. Notably, 87.9% (29 of 33) of positive isolates were classified in *Acanthamoeba* group II.

## Conclusion

This study revealed that *Acanthamoeba* spp. belonged to T4, which is the genotype commonly present in the eye of keratitis patients and had been isolated from cosmetic contact lenses and contact lens solution. These results support the need to improve medical knowledge of contact lens wearers about their proper care, ophthalmologist instructions, hygiene practice, and risk of nontrust or homemade cleaning lens solutions.

## Supporting information

**S1 File.**
(PDF)

**S2 File.**
(PDF)

## Author Contributions

**Conceptualization:** M. E. M. Tolba.

**Data curation:** Sara S. Abdel-Hakeem.

**Formal analysis:** Sara S. Abdel-Hakeem.

**Investigation:** Faten A. M. Hassan.

**Methodology:** Faten A. M. Hassan.

**Software:** Sara S. Abdel-Hakeem.

**Supervision:** M. E. M. Tolba, Gamal H. Abed, H. M. Omar, Sara S. Abdel-Hakeem.

**Visualization:** Faten A. M. Hassan.

**Writing – original draft:** Sara S. Abdel-Hakeem.

**Writing – review & editing:** M. E. M. Tolba, Gamal H. Abed, H. M. Omar.

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
