## [Decision Letter · Decision Letter 0]

4 Aug 2021

PONE-D-21-09515

Contact lenses contamination by Acanthamoeba spp. in Upper Egypt

PLOS ONE

Dear Miss Faten Abdo Hassan,

Thank you for submitting your manuscript to PLOS ONE. After careful consideration, we feel that it has merit but does not fully meet PLOS ONE’s publication criteria as it currently stands. Therefore, we invite you to submit a revised version of the manuscript that addresses the points raised during the review process.

We look forward to receiving your revised manuscript.

Kind regards,

Mohammad Mehdi Feizabadi, phd

Academic Editor

PLOS ONE

Journal Requirements:

Reviewers' comments:

Reviewer's Responses to Questions

**Comments to the Author**

1. Is the manuscript technically sound, and do the data support the conclusions?

Reviewer #1: Yes

Reviewer #2: Yes

2. Has the statistical analysis been performed appropriately and rigorously? 

Reviewer #1: Yes

Reviewer #2: I Don't Know

3. Have the authors made all data underlying the findings in their manuscript fully available?

Reviewer #1: Yes

Reviewer #2: Yes

4. Is the manuscript presented in an intelligible fashion and written in standard English?

Reviewer #1: Yes

Reviewer #2: Yes

5. Review Comments to the Author

Reviewer #1: - What kind of contact lens were assayed contamination by acanthamoebae?

- Were the lense solution related to the used lenses or new lenses or none of them ?

- Why the number of new lenses is not considered as the same as used lenses?

Reviewer #2: Thank you for providing me the opportunity to review this work. For this manuscript, the topic is a very good one, it fills a research gap nicely and is an area that I specialize in.

I recommend: Accept manuscript in its current form.

In my opinion, this paper has the potential to be accepted.

6. PLOS authors have the option to publish the peer review history of their article (what does this mean?). If published, this will include your full peer review and any attached files.

Reviewer #1: **Yes: **Abedinifar zohreh, Ph.D in Medical Microbiology

Reviewer #2: No

---

## [Author Response · Author response to Decision Letter 0]

17 Aug 2021

Original comments of the reviewer

1- What kind of contact lens were assayed contamination by Acanthamoeba?

2- Were the lenses solution related to the used lenses or new lenses or none of them?

3- Why the number of new lenses is not considered as the same as used lenses?

Reply by the author(s)

1-Cosmetic (colored, not intended for medical use) lenses were assayed. Manufacture names were not recorded to avoid legal sequences. 

2-lens solutions were related to use lenses.

3-New samples were easier to obtain and the surprising results that they might be contaminated made us explore more companies.

---

## [Editor Report · Decision Letter 1]

28 Oct 2021

Contact lenses contamination by Acanthamoeba spp. in Upper Egypt

PONE-D-21-09515R1

Dear Dr. 

Faten Abdo Hassan

We’re pleased to inform you that your manuscript has been judged scientifically suitable for publication and will be formally accepted for publication once it meets all outstanding technical requirements.

Kind regards,

Mohammad Mehdi Feizabadi, PhD

Academic Editor

PLOS ONE
---

## [Editor Report · Acceptance letter]

2 Nov 2021

PONE-D-21-09515R1 

Contact lenses contamination by Acanthamoeba spp. in Upper Egypt 

Dear Dr. Hassan:

I'm pleased to inform you that your manuscript has been deemed suitable for publication in PLOS ONE. Congratulations! Your manuscript is now with our production department. 

Kind regards, 

on behalf of

Dr. Mohammad Mehdi Feizabadi 

Academic Editor

PLOS ONE